# Weight Changes Are Linked to Adipose Tissue Genes in Overweight Women with Polycystic Ovary Syndrome

**DOI:** 10.3390/ijms252111566

**Published:** 2024-10-28

**Authors:** Anton Hellberg, Daniel Salamon, Dorina Ujvari, Mikael Rydén, Angelica Lindén Hirschberg

**Affiliations:** 1Department of Women’s and Children’s Health, Karolinska Institutet, 17177 Stockholm, Sweden; daniel.salamon@ki.se (D.S.); dorina.ujvari@ki.se (D.U.); angelica.hirschberg.linden@ki.se (A.L.H.); 2Department of Microbiology, Tumor and Cell Biology, National Pandemic Centre, Centre for Translational Microbiome Research, Karolinska Institutet, 17165 Solna, Sweden; 3Department of Medicine Huddinge (H7), Karolinska Institutet, C2-94 Karolinska University Hospital Huddinge, 14186 Stockholm, Sweden; mikael.ryden@ki.se; 4Department of Gynecology and Reproductive Medicine, Karolinska University Hospital Solna, 17176 Stockholm, Sweden

**Keywords:** PCOS, obesity, lifestyle, adipose tissue, gene expression, weight loss

## Abstract

Women with polycystic ovary syndrome (PCOS) have varying difficulties in achieving weight loss by lifestyle intervention, which may depend on adipose tissue metabolism. The objective was to study baseline subcutaneous adipose tissue gene expression as a prediction of weight loss by lifestyle intervention in obese/overweight women with PCOS. This is a secondary analysis of a randomized controlled trial where women with PCOS, aged 18–40 and body mass index (BMI) ≥ 27 were initially randomized to either a 4-month behavioral modification program or minimal intervention according to standard care. Baseline subcutaneous adipose tissue gene expression was related to weight change after the lifestyle intervention. A total of 55 obese/overweight women provided subcutaneous adipose samples at study entry. Weight loss was significant after behavioral modification (−2.2%, *p* = 0.0014), while there was no significant weight loss in the control group (−1.1%, *p* = 0.12). In microarray analysis of adipose samples, expression of 40 genes differed significantly between subgroups of those with the greatest weight loss or weight gain. 10 genes were involved in metabolic pathways including glutathione metabolism, gluconeogenesis, and pyruvate metabolism. Results were confirmed by real-time polymerase chain reaction (RT-PCR) in all 55 subjects. Expressions of *GSTM5*, *ANLN*, and *H3C2* correlated with weight change (R = −0.41, *p* = 0.002; R = −0.31, *p* = 0.023 and R = −0.32, *p* = 0.016, respectively). *GSTM5*, involved in glutathione metabolism, was the strongest predictor of weight loss, and together with baseline waist-hip ratio (WHR) explained 31% of the variation in body weight change. This study shows that baseline subcutaneous adipose tissue genes play a role for body weight outcome in response to lifestyle intervention in overweight/obese women with PCOS.

## 1. Introduction

Polycystic ovary syndrome (PCOS) is the most common endocrinopathy in women of reproductive age, with a prevalence of about 10% depending on the criteria used. The syndrome is characterized by oligo- or anovulation, polycystic ovarian morphology, and clinical or biochemical signs of hyperandrogenism [1]. The most frequent symptoms are menstrual irregularities, infertility, and hirsutism [2]. PCOS is also associated with obesity and insulin resistance (IR), as well as a long-term increased risk of type 2 diabetes (T2D) and metabolic syndrome [3,4,5]. Obesity and IR are known to exacerbate all classic symptoms of PCOS [6]. In addition, women with PCOS report that obesity causes significant anxiety and reduced quality of life [7,8].

Lifestyle intervention, including any combination of exercise, diet, and behavioral modification intervention, is the recommended first-line management for women with PCOS [1]. Several studies have shown that a minor weight loss of up to 5% can restore menstrual cyclicity and ovulation and also improve metabolic variables [1,9,10]. However, a great clinical problem is posed by many patients failing to lose weight by lifestyle changes or quickly regaining weight after an initial weight loss [10,11,12]. Women with PCOS may even have more difficulty succeeding in weight loss regimens compared to other women [13]. For resumption of ovulation, a short-term lifestyle intervention may be sufficient, but for metabolic improvement, continued weight maintenance is necessary [1,10]. It is important to identify factors predicting successful weight loss and maintenance of lower body weight by lifestyle intervention.

PCOS is associated with several aspects of adipose tissue dysfunction, including impaired insulin signaling and glucose transport, adipokine dysregulation, chronic inflammation, and oxidative stress, as well as dysregulated lipolysis [14]. Subcutaneous lipolysis is important for body weight regulation and risk of weight gain, and thereby the development of IR and T2D [15]. Specifically, it has been demonstrated that expression of adipose tissue lipolysis-regulating genes like *PRKAR2B*, *MGLL*, *FABP4*, and *AQP7* can predict future weight gain [15]. In women with PCOS, increased catecholamine-induced lipolysis in visceral fat but decreased lipolysis in subcutaneous fat cells has been reported [16,17]. Furthermore, hypertrophic adipocytes are associated with IR, T2D, and cardiovascular disease, and studies on women with PCOS have shown larger adipocytes in abdominal subcutaneous tissue independent of body mass index (BMI) [17,18]. However, the role of adipocyte size and subcutaneous adipose tissue gene expression for body weight regulation in response to lifestyle intervention in women with PCOS has not been explored.

The present study is based on our prior randomized controlled behavioral modification intervention in overweight/obese women with PCOS [10]. The aim was to investigate the expression of baseline subcutaneous adipose tissue metabolic regulating genes as prediction of weight loss by lifestyle intervention in these women.

## 2. Results

### 2.1. Clinical Characteristics of the Behavioral Intervention and Minimal Intervention Groups

Clinical characteristics at baseline were comparable between the two groups of women undergoing four months of behavioral modification intervention or minimal intervention (control treatment) (Table 1). In the behavioral modification group (*n* = 29), there were significant decreases in body weight, BMI, total fat percentage, trunk fat mass, total cholesterol, and low-density lipoprotein cholesterol after the intervention (Table 1). In the control treatment group (*n* = 26), there were significant decreases in testosterone, free androgen index, and aspartate transaminase (ASAT) but not in weight loss. There was no significant difference in weight change between the groups. In the behavioral modification group, 17% of the women succeeded in losing more than 5% of their body weight, while 21% even gained weight (Figure 1). The corresponding percentages in the control treatment group were 15% and 42%, respectively.

### 2.2. Clinical Characteristics of Subgroups of Weight Loss and Weight Gain for Microarray Analysis

Clinical characteristics of subgroups with the most weight loss (*n* = 5) and the most weight gain (*n* = 5) (regardless of what treatment arm they belonged to) at baseline and after four months of lifestyle intervention are provided in Table 2. After intervention, the subgroups differed significantly in the expected direction regarding body weight (*p* < 0.01), BMI (*p* < 0.01), total fat mass (*p* < 0.05), total trunk fat mass (*p* < 0.05), and homeostatic model assessment for insulin resistance (HOMA-IR) (*p* < 0.05). There was no significant difference in average adipocyte size within or between the subgroups.

### 2.3. Microarray and Pathway Analysis of the Subgroups of Weight Loss and Weight Gain

Table 3 shows the results of all gene expressions that were significantly different between the subgroups of weight loss (*n* = 5) and weight gain (*n* = 5) by microarray analysis. Gene expressions are described as a ratio between the Subgroup of weight loss and the Subgroup of weight gain. Further pathway analysis showed that gene expressions of *GSTM5*, *RRM2*, *ANLN*, *ANPEP*, *TOP2A*, *STMN1;MIR3917*, *H3C2*, *PFKB1*, *ACLY*, and *PC* were connected in different metabolic pathways, including glutathione metabolism, gluconeogenesis, pyruvate metabolism, and the citrate cycle, as marked in bold in Table 3. Two of the genes connected to metabolic pathways were also associated with retinoblastoma cancer genes, and two genes were connected to retinoblastoma cancer genes only.

### 2.4. Real-Time PCR Analysis of All Individuals

To confirm the results from the microarray and pathway analysis, all individuals who completed the four-month trial and provided baseline adipose tissue samples (*n* = 55) had their tissue samples analyzed by real-time polymerase chain reaction (RT-PCR). The selected genes were from the results of microarray and pathway analysis and those connected to different metabolic pathways (*n* = 10). The whole sample (*n* = 55) was divided into two groups, regardless of lifestyle treatment arm, based on the 50th centile of their weight change from baseline. The Weight loss group (*n* = 28) with a mean weight loss of −4.34% (−5.35–−3.33), (*p* < 0.001) and the Weight gain group (*n* = 27) with a mean weight gain of 1.12% (0.19–2.04), (*p* = 0.026). Figure 2a–j shows the differences in relative expression of the 10 genes between the Weight loss group and the Weight gain group for the whole sample. The relative expression of *GSTM5* and *H3C2* were significantly higher in the Weight loss group compared to the Weight gain group, and the expression of *RRM2* and *ANLN* tended to be higher in the Weight loss group (*p* = 0.051 and *p* = 0.056, respectively).
ijms-25-11566-t003_Table 3Table 3Gene ID and encoded protein for microarray analysis, ratio of gene expressions between the subgroups of weight loss (*n* = 5) and weight gain (*n* = 5) and *p*-values for the difference between groups, FDR for each Gene ID and pathway for Gene IDs.Gene IDEncoding ProteinWeight Loss: Weight Gain Ratio*p*-ValueFDRPathway***GSTM5*****Glutathione S-transferase mu 5****1.32****0.0005****0.2498****Glutathione metabolism, Glutathione-mediated detoxification***PARVG*Parvin, gamma1.292.90 × 10^−5^0.0811
*ARHGAP11B*; *ARHGAP11A*Rho GTPase activating protein 11B; Rho GTPase activating protein 11A1.210.00020.1836
*MSR1*Macrophage scavenger receptor 11.200.00040.223
*ALCAM*Activated leukocyte cell adhesion molecule1.195.38 × 10^−5^0.0962
*CPXM1*Carboxypeptidase X (M14 family), member 11.180.00020.1621
***RRM2*****Ribonucleotide reductase M2****1.16****6.41 × 10^−6^****0.0603****Glutathione metabolism, Glutathione-mediated detoxification, Retinoblastoma gene in cancer***CD52*CD52 molecule1.163.43 × 10^−5^0.0811
*SIPA1L2*Signal-induced proliferation-associated 1 like 21.140.00020.1836
***ANLN*****Anillin actin binding protein****1.12****8.20 × 10^−6^****0.0603****Retinoblastoma gene in cancer***GPR183*G protein-coupled receptor 1831.120.00030.1877
*CDK15*Cyclin-dependent kinase 151.120.00040.2233
*H2BC14*Histone cluster 1, H2bm1.118.43 × 10^−6^0.0603
*H2AC11*Histone cluster 1, H2ag1.090.00030.1877
***ANPEP*****Alanyl (membrane) aminopeptidase****1.07****2.66 × 10^−5^****0.0811****Glutathione metabolism*****TOP2A*****Topoisomerase (DNA) II alpha****1.07****0.0002****0.1551****Retinoblastoma gene in cancer, DNA replication***IL1RN*Interleukin 1 receptor antagonist1.064.97 × 10^−5^0.0962
***STMN1; MIR3917*****Stathmin 1; microRNA 3917****1.06****0.0001****0.1551****Retinoblastoma gene in cancer***H1-5*Histone cluster 1, H1b1.052.36 × 10^−5^0.0811
*CD83*CD83 molecule1.050.00010.1548
***H3C2*****Histone cluster 1, H3b****1.02****0.0002****0.1637****DNA replication***TM7SF2*Transmembrane 7 superfamily member 20.970.00030.1983
*THRSP*Thyroid hormone responsive0.969.32 × 10^−5^0.1333
***PFKFB1*****6-phosphofructo-2-kinase/fructose-2,6-biphosphatase 1****0.96****0.0001****0.1548****Gluconeogenesis***APOL1*Apolipoprotein L10.953.46 × 10^−5^0.0811
*GBP4*Guanylate binding protein 40.953.78 × 10^−5^0.0811
*MARC1*Mitochondrial amidoxime reducing component 10.950.00010.1548
*SLC4A4*Solute carrier family 4, member 40.950.00030.1983
*LGALS12*Lectin, galactoside-binding, soluble, 120.950.00040.2233
*PTPRU*Protein tyrosine phosphatase, receptor type, U0.940.00030.1877
*GBP1*Guanylate binding protein 1, interferon-inducible0.939.24 × 10^−5^0.1333
*SLC19A3*Solute carrier family 19, member 30.911.26 × 10^−5^0.0678
*TMEM246*Transmembrane protein 2460.900.00020.1836
*SLC25A26*Solute carrier family 25, member 260.890.00020.1776
***ACLY*****ATP citrate lyase****0.88****0.0002****0.1836****Pyruvate metabolism, Citrate cycle***MAMLD1*Mastermind-like domain containing 10.870.00040.223
*PLEKHA6*Pleckstrin homology domain containing, family A member 60.870.00050.2498
***PC*****Pyruvate carboxylase****0.86****0.0003****0.1877****Gluconeogenesis, Pyruvate metabolism, Citrate cycle***DEFB1*Defensin, beta 10.810.00010.1548
*SLC7A10*Solute carrier family 7, member 100.796.46 × 10^−5^0.1066
Genes involved in pathways allocated from the pathway analysis are provided in bold. FDR, False Discovery Rate.

### 2.5. Correlations Between Baseline Gene Expression and Weight Change

In the whole sample (*n* = 55), baseline gene expression of *GSTM5*, *ANLN* and *H3C2* correlated inversely to weight change after lifestyle intervention; R = −0.41, *p* = 0.0017; R = −0.31, *p* = 0.023 and R = −0.32, *p* = 0.016, respectively (Figure 3a–d). *RRM2* correlated negatively but the correlation did not reach a level of significance (R = −0.2, *p* = 0.15) (Figure 3a–d). Baseline gene expression of *GSTM5* also tended to correlate with average adipocyte size at baseline (R = 0.25, *p* = 0.055). There was no correlation between baseline weight and weight change (R = −0.09, *p* = 0.52).

### 2.6. Multiple Regression Analysis

Of the 10 genes, expression of *GSTM5* and *H3C2* explained 27% of the variation in body weight change measured in percentage (*p* < 0.001) in the multiple regression analysis of the whole sample (*n* = 55). The strongest predictor of weight loss was *GSTM5,* explaining 19% of change in weight measured in percentage (*p* < 0.001), *H3C2* explained 10.3% of the weight change (*p* = 0.02). When including also clinical variables in the model, the multiple regression analysis revealed that baseline expression of *GSTM5* together with baseline (waist-hip ratio) WHR explained 31% of the variation in body weight change (*p* < 0.001) (Table 4), of which *GSTM5* was the strongest predictor explaining 18.2% of the variation (*p* = 0.0014). *GSTM5* and WHR had an AUC of 0.729 (95% CI: 0.571–0.887) (Figure 4).

## 3. Discussion

In this study, we have shown the influence of baseline subcutaneous adipose tissue gene expression for body weight outcome after lifestyle intervention in overweight/obese women with PCOS. Among the expression of 40 genes that differed significantly in the microarray analysis between the subgroups of weight loss and weight gain after participating in a lifestyle intervention study, 10 genes were found after pathway analysis to be involved in various metabolic pathways, such as glutathione metabolism (*GSTM5*, *RRM2*, *ANPEP*), gluconeogenesis (*PFKB1* and *PC*), citrate cycle (*ACLY* and *PC*), pyruvate metabolism (*ACLY* and *PC*), and DNA replication (*H3C2*). After validation with RT-PCR of the whole sample, the relative baseline expression of *GSTM5*, *ANLN*, and *H3C2* correlated with weight change after lifestyle intervention. The strongest predictor of weight change among all genes involved in metabolic pathways and clinical variables studied was the relative expression of *GSTM5* involved in glutathione metabolism.

There is evidence that lifestyle intervention improves reproductive and metabolic health in women with PCOS [1,9,10]. A healthy lifestyle is therefore recommended for all women with PCOS to optimize general health, quality of life, body composition, and weight control to prevent weight gain or to lose weight in those with overweight/obesity [1]. However, studies show that the success rate of a lifestyle intervention varies between individuals [11,12], and failure to adhere to dietary, exercise, and/or behavioral programs can be debilitating and cause additional stigma for these patients. Failure to lose weight after lifestyle intervention is likely multifactorial. One reason for the lack of response could be genetic constitution [11]. We have also previously shown that psychological well-being and personality factors could impact successful weight loss [7]. The present study furthermore supports that the expression of certain genes in subcutaneous adipose tissue, particularly *GSTM5*, can influence weight outcome in response to lifestyle intervention in women with PCOS.

*GSTM5* belongs to the glutathione transferase (GST) superfamily (Alpha, Mu, Pi, Sigma, Theta, and Zeta) and to the mu-class consisting of 5 enzymes and is involved in cell metabolism and the metabolic detoxification of several endogenous and exogenous compounds, among them reactive oxygen species (ROS), electrophilic compounds, carcinogens, and medications by conjugation with glutathione [19,20,21,22]. Genes belonging to the GST superfamily have been studied in metabolic disorders; for example, *GSTA4* has been shown to be downregulated in obese individuals in relation to the inflammatory state associated with obesity [21,23], and *GSTM1* null-polymorphisms have been linked to coronary heart disease [24]. Furthermore, a randomized clinical trial (RCT) of glutathione supplementation demonstrated improved IR and decreased body fat percentage and fatty liver in older individuals with glutathione deficiency [25].

The clinical role of GSTMs is not elucidated [19], and particularly not in PCOS. One previous study in non-obese adolescent girls with PCOS demonstrated that carriers of the *GSTM1*-null genotype have significantly lower levels of testosterone in comparison to PCOS carriers of the *GSTM1*-active genotype, which may be related to the ability of *GSTM1* to serve as a steroid-binding protein of testosterone [26]. The null-polymorphisms of *GSTT1* and *GSTM1* have also, in combination, but not separately, been associated in the etiology of PCOS [27]. A study on ovarian cancer reported decreased *GSTM5* expression in cancer tissue compared to normal tissue, and the expression was positively correlated with ovarian cancer prognosis [28]. Data also indicate that *GSTM5* expression might reduce ROS levels to ameliorate oxidative stress. It can be speculated that *GSTM5* deficiency in adipose tissue related to failure to lose weight in PCOS, as shown in the present study, may be linked to low-grade chronic inflammation, oxidative stress, and insulin resistance [14,29]. In addition, animal studies lend support for a role of glutathione in counteracting obesity by enhancing insulin sensitivity and promoting lipid degradation [30].

It is well known that chronic inflammation and oxidative stress have a significant role in obesity [31]. In PCOS, adipocyte dysfunction, independently of obesity, is associated with systemic chronic inflammation and oxidative stress, in turn linked to clinically adverse cardiometabolic profiles [32,33,34]. Furthermore, oxidative stress and increased production of ROS may contribute to infertility and miscarriage in women with PCOS, and oxidative stress has also been associated with miscarriage in PCOS-like rats [35]. However, more research is needed to understand the clinical role of adipocyte dysfunction in PCOS [14]. This report lends support for a role of *GSTM5* as a target for predicting successful weight loss through lifestyle intervention.

Some limitations should be pointed out. In this study, we collected subcutaneous adipose tissue biopsies, whereas visceral adipose tissue is associated with a more detrimental metabolic profile [16,36]. However, visceral adipose tissue biopsies are difficult to collect, requiring invasive surgical interventions. Importantly, adipose tissue dysfunction has also been demonstrated in subcutaneous adipose tissue in women with PCOS [1,17]. For the detection of gene expression, a false discovery rate of <0.25 was used. This value was, after consultation with a bioinformatician, considered acceptable in an explorative study as the present one [37]. In support, several genes were found to differ in relative gene expression in the RT-PCR analysis for the various comparisons in relation to weight change after lifestyle intervention. The reason why not all genes were significantly different between the groups of weight loss and weight gain might be explained by limited power to detect significant differences.

The strengths of the study include a well-characterized population of women fulfilling all three criteria of PCOS, a relatively large number of participants providing adipose tissue biopsies at baseline, evaluation of weight outcome after a structured behavioral modification program as part of an RCT, and robust validation with RT-PCR and pathway analysis of the microarray results.

In conclusion, we have demonstrated that subcutaneous adipose tissue gene expressions are of importance for prediction of weight loss in response to participation in a lifestyle intervention study in overweight/obese women with PCOS. Among the metabolic-regulating genes studied, the gene expression of *GSTM5*, a gene involved in glutathione metabolism, was the strongest predictor of weight loss in these women. Our results of different expression of subcutaneous adipose tissue genes of importance for body weight regulation may contribute to the understanding of the large variation in weight change following lifestyle intervention in women with PCOS. Hopefully, increased knowledge about how to identify which women with PCOS benefit most from lifestyle programs can guide doctors to offer more individualized treatment in the future.

## 4. Materials and Methods

### 4.1. Participants and Study Design

Overweight or obese women with PCOS (*n* = 68) participated in an RCT conducted at the Karolinska University Hospital, Stockholm, Sweden, comparing behavioral modification intervention with minimal intervention for four months as previously described by Oberg et al. [10]. The inclusion criteria were: 18–40 years of age, BMI ≥ 27, and all 3 of the Rotterdam diagnostic criteria of PCOS (phenotype A of PCOS) [38], i.e., oligomenorrhea or amenorrhea, polycystic ovaries on a transvaginal ultrasound scan and displaying clinical (hirsutism, acne, or androgenic alopecia) or biochemical (elevated serum androgen level) hyperandrogenism. Exclusion criteria were pregnancy or breastfeeding, history of eating disorder, substantial weight change during the past year, smoking, taking regular medication, and another medical condition including congenital adrenal hyperplasia, Cushing’s syndrome, thyroid dysfunction, hyperprolactinemia, and virilizing tumor.

Women were randomized in a ratio of 1:1 to either receive behavioral modification intervention (*n* = 34) or minimal intervention (control treatment) (*n* = 34) for four months. The behavioral modification treatment consisted of a structured approach for long-term weight control to improve reproductive and metabolic function [10]. The treatment was characterized by a formal course in small groups, held 3 times a month, and led by a lifestyle coach with a PhD in endocrinology and metabolism. The course included knowledge concerning weight control, personal leadership, mindfulness, physical activity, and diet and contained reading material, homework, group discussions, and personalized meetings to discuss individual training regimes, diet changes, and to ensure compliance.

The minimal intervention, designed to reflect standard patient care, consisted of recommendations given by a research midwife regarding a general healthy lifestyle supported by a pamphlet with written advice about diet and exercise. Women in both groups attended monthly visits to measure weight, waist, hip, and vital signs. The participants were encouraged to wear an accelerometer (ActiGraph GT3X) for seven days at baseline and at the 4-month follow-up for measurements of energy expenditure.

The women were assessed at baseline, 4 months, and 12 month follow-up on menstrual cycle days 6–8 after spontaneous or induced menstrual bleeding by progestogen. Gynecological examination and anthropometric measurements (height, weight, hip, and waist circumference) were performed, and fasting venous blood samples were collected for analysis of hormones and metabolic parameters. Body weight was measured on the same electronic scale for all women wearing underwear and a light hospital shirt. Analytical methods for hormones, binding proteins, and metabolic factors are previously reported [10]. The participants also underwent assessment of body composition by Dual-Energy X-ray Absorptiometry (DEXA) scanner Hologic Discovery A, manufactured by Hologic, and collection of biopsies of the endometrium, muscle, and subcutaneous adipose tissue. The subcutaneous adipose tissue samples were collected under standard sampling technique from the abdomen at the level between the umbilicus and the iliac crest through a small incision in the skin under local anesthesia and snap frozen and stored at −80 °C.

The Regional Review Board of Ethics in Research in Stockholm approved the study protocol (2012/146–31/3). The trial was registered with the clinical trial registry number: ISRCTN48947168.

### 4.2. Procedure of the Present Study

Out of 57 individuals completing the 4-month trial, 55 provided baseline adipose tissue samples, and these women constitute the material of the present study, Figure 5. Of these individuals, regardless of what treatment arm they belonged to, 5 were selected that had lost the most weight and 5 that had gained the most weight for analysis of adipose tissue gene expression by microarray after total RNA extraction. The gene expressions that differed significantly between the subgroups in the microarray analysis were processed in a gene list enrichment analysis for metabolic pathways. Finally, total RNA extraction, cDNA synthesis and RT-PCR were performed on the whole study material (*n* = 55) to validate the results. For comparison, the whole material was divided into 2 groups based on the 50th centile of their weight change: a Weight loss group and a Weight gain group.

### 4.3. RNA Extraction

Frozen adipose tissue was homogenized in 1 mL Trizol for microarray samples and Qiazol for whole sample using Tissuelyser (Qiagen, Hilden, Germany) with settings 2 × 2 min/25.0 for RNA extraction. A standard Trizol protocol was utilized, RNA cleanup was performed with Qiagen RNeasy mini kit (Qiagen, Hilden, Germany). Quality control of total RNA was performed with Agilent Technologies 2200 Tapestation (Agilent, Santa Clara, CA, USA). NanoDrop ND-1000 Spectrophotometer (Thermo Scientific, Wilmington, NC, USA) was used for measurement of RNA concentration.

### 4.4. Microarray

Biotinylated DNA targets were prepared from 100 ng total RNA using the GeneChip WT Plus Reagent Kit (Thermo Scientific, Wilmington, NC, USA) according to the manufacturer’s instructions. Hybridization, washing, and staining were carried out on Affymetrix Clariom S, human arrays, using Affymetrix GeneChip^®^ Fluidics Station 450 (Thermo Scientific, Wilmington, NC, USA), according to the manufacturer’s protocol. The fluorescent intensities were determined with Affymetrix GeneChip Scanner 3000 7 G (Thermo Scientific, Wilmington, NC, USA).

### 4.5. Pathway Analysis

The genes found to differ in baseline expression between the subgroups of weight loss and weight gain were processed in a gene list functional enrichment analysis for metabolic pathways, the ToppGene function ToppFun [39]. A *p* < 0.05 was set as the cut-off limit for genes connected in the same pathway.

### 4.6. RT-PCR

SuperScript VILO cDNA Synthesis Kit (Thermo Fisher Scientific, Waltham, MA, USA) was used for cDNA synthesis and the SybrGreen method for determination of gene expression levels. To normalize gene expression levels, ribosomal protein L13A (RPL13A) was used as a housekeeping gene. Determination of the relative gene expression levels was performed with the ΔΔCt method. All reactions were run in triplicates. RT-PCR was performed on StepOnePlus Real-time PCR systems (Thermo Fisher Scientific, USA). Appendix A lists the applied oligonucleotides (Sigma-Aldrich, St. Louis, MO, USA).

### 4.7. Adipocyte Morphology

Fixed tissue samples were embedded in paraffin and stained with hematoxylin and eosin. Using CellInsight CX5 High Content Screening (HCS) Platform (IC904000)(4× magnification) photomicrographs were taken, 4 images per fixed sample. The methods have been described in detail previously [40]. Average adipocyte size (Feret’s diameter) was calculated using ImageJ (version 1.52) and the plugin Adipocyte Tools [41].

### 4.8. Statistical Analysis

Statistical analyses were performed using the RStudio statistical program (version 1.4.1106) and Statistica TIBCO Software Inc., Santa Clara, CA, USA, 241027. (version 14.0). Tests for normality were assessed using Shapiro–Wilk test, test of skew, visual scrutiny of histograms and QQ-plots. Mixed-model analysis of variance (mixed-model ANOVA), with the factors Group, Time and the interaction Group*Time, was used to analyze within and between-group differences for behavioral modification intervention and minimal intervention. Values for the subgroups are presented as median and interquartile range (25th–75th). With-in group differences were calculated using subtraction of the median values of the 2 time-points, and between group differences with Mann–Whitney U test. A *p* < 0.05 and a false discovery rate < 0.25 were set to detect significant differences in gene expression in the microarray analysis. Gene expression ratios of the weight loss and weight gain subgroups were calculated by dividing the results of these two subgroups. All gene expressions of the RT-PCR analysis were logarithmically transformed. Logarithmically transformed variables are presented untransformed.

Furthermore, all individuals who had completed the 4-month trial and provided adipose tissue samples at baseline (*n* = 55) were divided into two groups based on the 50th centile of their weight change, regardless of lifestyle treatment arm. The relative gene expression of the Weight loss group and the Weight gain group was compared using one-way analysis of variance (one-way ANOVA). Extremes were removed before one-way ANOVA if the assumption of equal variance was not met. Spearman or Pearson’s correlations, based upon homoscedasticity, were performed to study associations between selected gene expression and weight change in the whole study group. Multiple regression analysis was used to evaluate which of the baseline variables was the strongest predictor of weight change. An area under the curve and receiver operating characteristic (ROC) curve were performed on the strongest predictors of weight loss (%) as a binary variable with 0% as the cutoff. Overall, a *p* < 0.05 was considered significant.

## Figures and Tables

**Figure 1 ijms-25-11566-f001:**
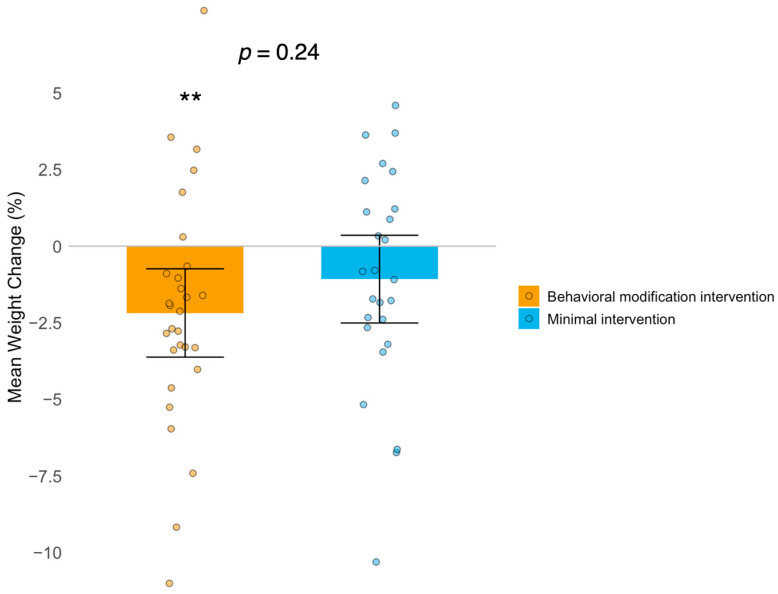
Weight change (%) after 4 months of intervention in the behavioral modification group and the minimal intervention group (control treatment). Results presented as mean change from baseline and confidence interval. Body weight decreased significantly (*p* = 0.0014, represented by **) in the behavioral modification group but not in the control treatment group. There was no significant difference between the two treatment groups (*p* = 0.24).

**Figure 2 ijms-25-11566-f002:**
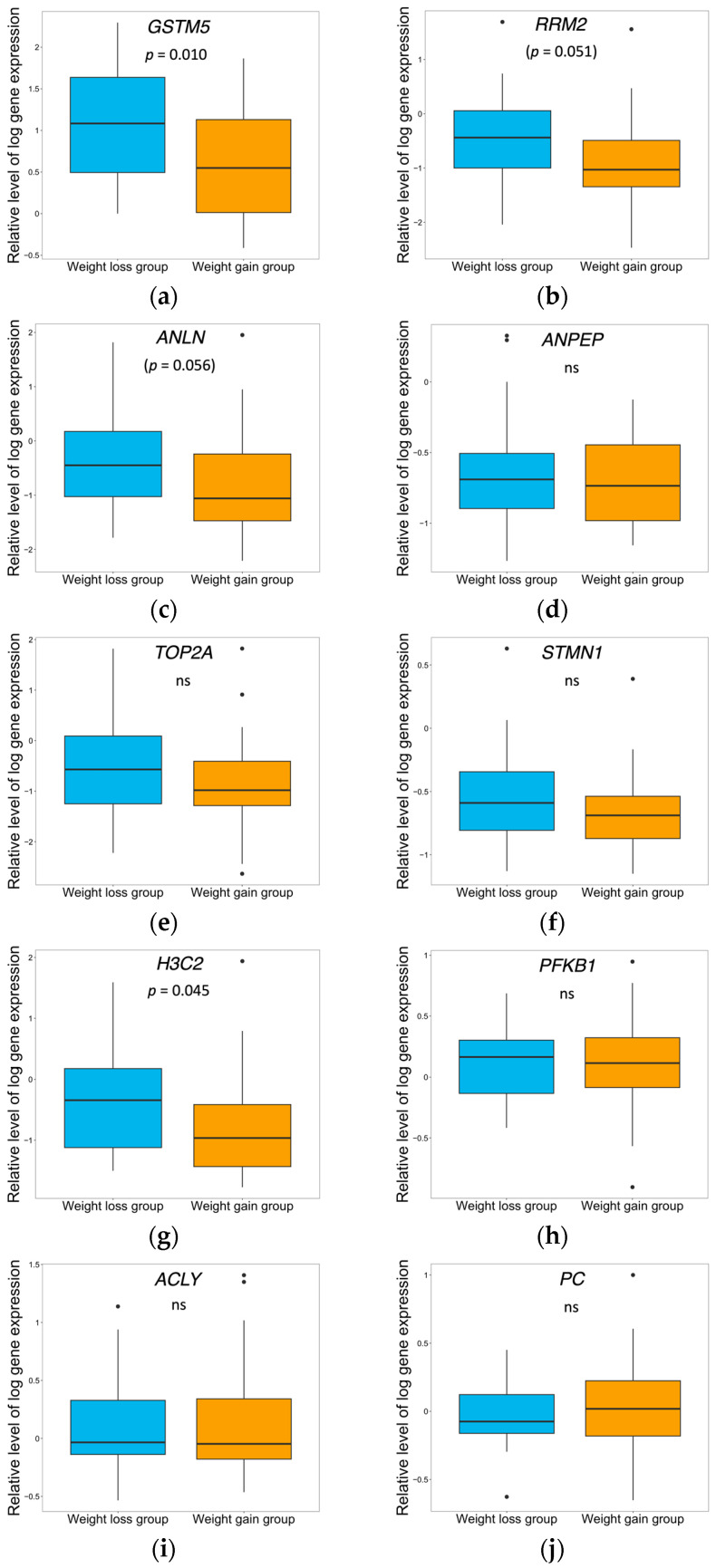
(**a**–**j**). Relative expression of (**a**) *GSTM5*, (**b**) *RRM2*, (**c**) *ANLN*, (**d**) *ANPEP*, (**e**) *TOP2A*, (**f**) *STMN1;MIR3917*, (**g**) *H3C2*, (**h**) *PFKB1*, (**i**) *ACLY*, and (**j**) *PC* of the whole sample (*n* = 55) divided into the 50th centile of weight change from baseline into a Weight loss group and a Weight gain group. (**a**) *GSTM5* and (**g**) *H3C2* differed significantly in relative expression between the groups, *p* = 0.01 and *p* = 0.045, respectively. Black dots represent extreme values. ns = Not significant.

**Figure 3 ijms-25-11566-f003:**
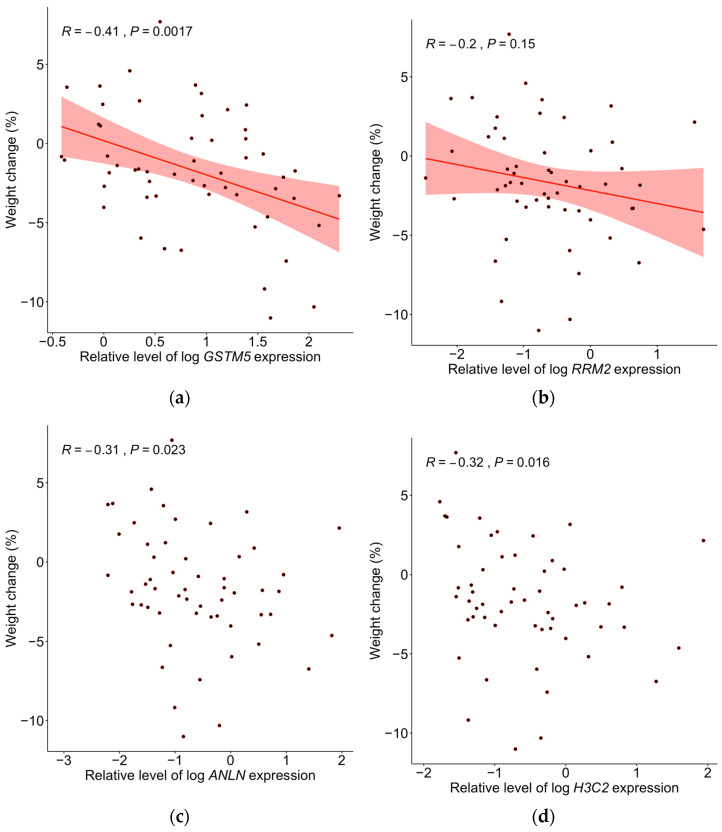
(**a**–**d**). Correlation between individual relative expression values of (**a**) *GSTM5*, (**b**) *RRM2*, (**c**) *ANLN* and (**d**) *H3C2* (*n* = 55) and change in weight (Weight Change) for the whole sample (*n* = 55). Pearson correlation with regression line and confidence intervals were used for statistical analysis of (**a**) *GSTM5* and (**b**) *RRM2*. Spearman correlations were used for statistical analysis of (**c**) *ANLN*, and (**d**) *H3C2*.

**Figure 4 ijms-25-11566-f004:**
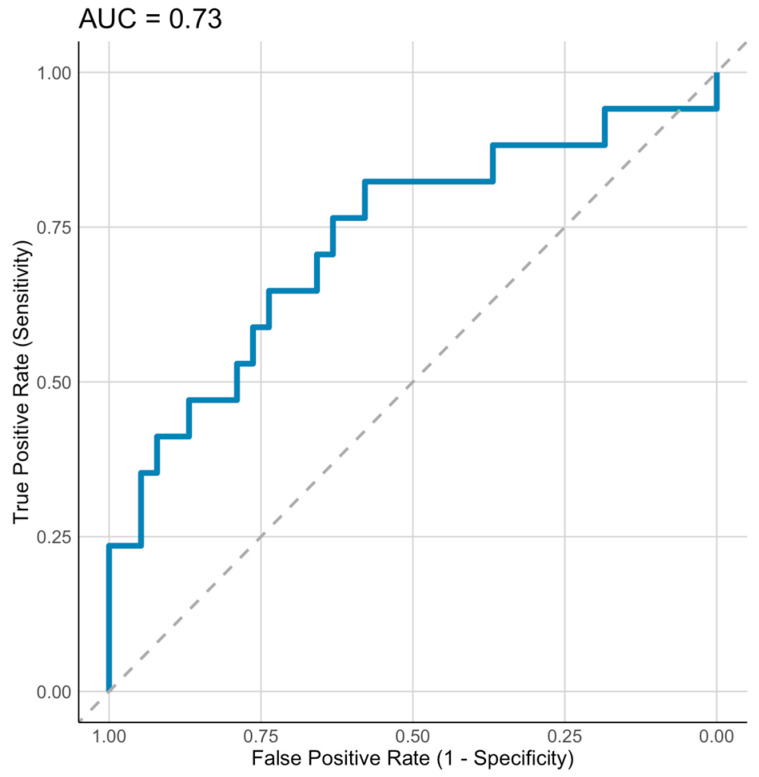
Receiver operating characteristic (ROC) curve of *GSTM5* and WHR for prediction of weight loss (%) in the whole sample (*n* = 55). Dashed line represents random line. Blue solid line represents fitted curve.

**Figure 5 ijms-25-11566-f005:**
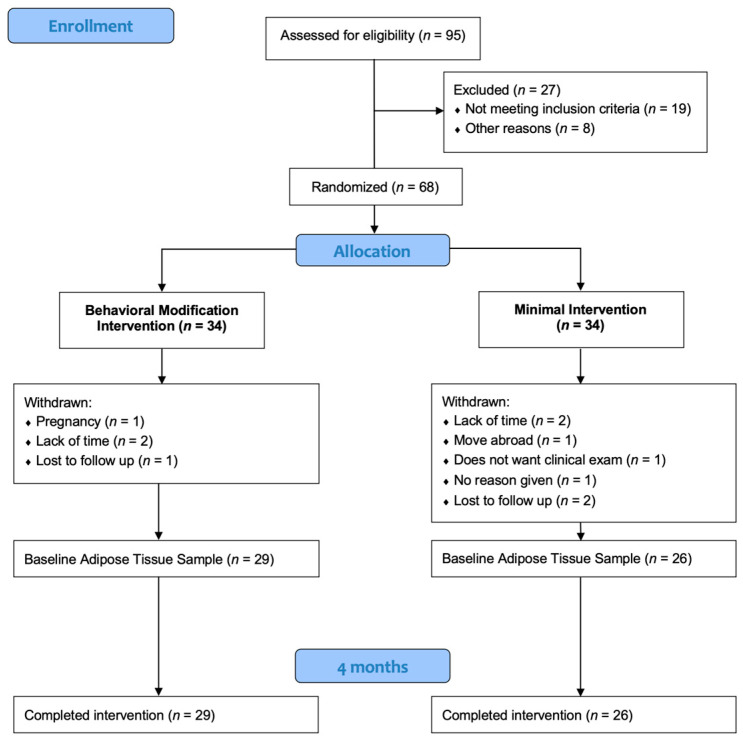
Diagram of the study design.

**Table 1 ijms-25-11566-t001:** Clinical characteristics of participants randomized to behavioral modification or minimal intervention.

	Behavioral Modification Intervention (*n* = 29)			Minimal Intervention (*n* = 26)			
Parameters	Baseline	4 Months	*p*-Value	Baseline	4 Months	*p*-Value	Between-Group Change
Age	31.1 (29.3–33.0)			29.9 (27.9–32.0)			
**Anthropometric**							
Bodyweight (kg)	93.4 (87.8–99.0)	91.3 (85.7–97.0)	**0.0014**	93.3 (87.7–98.9)	92.2 (86.6–97.8)	0.12	−1.07 (−2.88–0.75)
BMI	33.7 (31.9–35.4)	32.9 (31.2–34.6)	**0.0017**	34.0 (32.3–35.8)	33.6 (31.9–35.4)	0.12	−0.38 (−1.04–0.29)
WHR	0.90 (0.88–0.92)	0.90 (0.88–0.92)	0.98	0.88 (0.86–0.91)	0.88 (0.86–0.90)	0.83	0.002 (−0.03–0.02)
Total fat (%)	42.6 (41.2–44.0)	41.7 (40.3–43.1)	**0.0079**	43.5 (42.1–44.9)	43.3 (41.9–44.7)	0.67	−0.79 (−1.78–0.20)
Trunk fat mass (kg)	20.2 (18.3–22.1)	19.4 (17.5–21.3)	**0.004**	20.5 (18.6–22.3)	20.2 (18.3–22.1)	0.47	0.63 (−1.4–0.18)
Lean body mass (kg)	51.6 (48.8–54.4)	51.4 (48.6–54.2)	0.69	50.7 (48.0–53.5)	50.7 (47.8–53.5)	0.87	−0.1 (−1.6–1.3)
**Endocrine**							
FSH (IU/L) ^L^	6.80 (5.58–8.01)	6.90 (5.65–8.15)	0.45	5.46 (4.26–6.66)	5.86 (4.53–7.19)	0.64	−0.31 (−1.87–1.26)
LH (IU/L)	6.59 (5.10–8.08)	7.70 (6.14–9.26)	0.20	7.63 (6.16–9.10)	6.66 (4.95–8.37)	0.30	2.08 (−0.45–4.61)
Testosterone ^L^ (nmol/L)	1.23 (1.06–1.40)	1.19 (1.01–1.36)	0.47	1.42 (1.24–1.60)	1.28 (1.09–1.47)	**0.023**	0.10 (−0.09–0.29)
SHBG (nmol/L)	27.1 (22.0–32.2)	27.8 (22.6–32.9)	0.63	25.9 (20.8–31.0)	26.4 (21.0–31.7)	0.78	0.25 (−4.17–4.68)
FAI	5.59 (4.14–7.05)	5.40 (3.90–6.90)	0.76	7.60 (6.10–9.10)	5.78 (4.14–7.42)	**0.014**	1.62 (−0.30–3.55)
**Metabolic**							
HOMA-IR	3.39 (2.39–4.40)	3.12 (2.09–4.16)	0.40	3.33 (2.29–4.37)	3.54 (2.47–4.61)	0.57	−0.48 (−1.45–0.50)
Triglycerides (mmol/L)	1.25 (1.03–1.46)	1.12 (0.90–1.34)	0.20	1.35 (1.14–1.56)	1.16 (0.92–1.39)	0.07	0.07 (−0.21–0.35)
Cholesterol (mmol/L)	4.87 (4.61–5.14)	4.59 (4.31–4.87)	**0.013**	4.80 (4.53–5.07)	4.66 (4.37–4.96)	0.27	−0.15 (−0.48–0.17)
HDL (mmol/L)	0.97 (0.87–1.07)	1.03 (0.92–1.13)	0.13	1.08 (0.98–1.18)	1.16 (1.05–1.27)	0.07	−0.02 (−0.13–0.09)
LDL (mmol/L)	3.34 (3.10–3.58)	3.05 (2.80–3.29)	**0.002**	3.10 (2.86–3.34)	2.98 (2.72–3.24)	0.20	−0.17 (−0.43–0.09)
ASAT (mikrokat/L) ^L^	0.40 (0.35–0.45)	0.36 (0.31–0.42)	0.42	0.39 (0.34–0.45)	0.34 (0.28–0.40)	**0.045**	0.02 (−0.06–0.1)
ALAT (mikrokat/L) ^L^	0.24 (0.16–0.31)	0.23 (0.15–0.31)	0.56	0.27 (0.20–0.34)	0.20 (0.12–0.29)	0.26	0.06 (−0.09–0.20)
Adjusted adipocyte size (μm)	122 (120–124)	123 (120–125)	0.46	121 (119–123)	122 (119–124)	0.63	0.33 (−3.63–4.29)

Values presented as median interquartile range (25th–75th). Significant results presented in bold. ALAT, alanine transaminase; ASAT, aspartate transaminase; BMI, body mass index; FAI, free androgen index; FSH, follicle-stimulating hormone; HDL, high-density lipoprotein cholesterol; HOMA-IR, homeostatic model assessment for insulin resistance; LDL, low-density lipoprotein cholesterol; LH, luteinizing hormone; SHBG (sex hormone-binding globulin); WHR, waist-hip ratio. ^L^ Denotes log transformed variables.

**Table 2 ijms-25-11566-t002:** Clinical characteristics at baseline and after 4 months of lifestyle intervention in the weight loss and weight gain subgroups.

		Outlier Weight Loss Group (*n* = 5)		Outlier Weight Gain Group (*n* = 5)			*p*-Value
Parameters	Baseline	4 Months	Change	Baseline	4 Months	Change	Change Between Groups
Anthropometric							
Body weight (kg)	87.6 (80.6–88.8)	81.1 (73.2–83.5)	−5.3 (−6.5–−4.9)	82.7 (81.5–85.8)	86.5 (84.4–92.4)	3.4 (2.9–3.8)	**0.008**
BMI	31.8 (31.1–33.1)	29.4 (29.2–30.1)	−1.85 (−2.36–−1.70)	30.1 (29.6–31.1)	31.2 (30.4–32.3)	1.15 (1.07–1.31)	**0.008**
WHR	0.89 (0.89–0.89)	0.88 (0.85–0.89)	−0.03 (−0.04–0.00)	0.83 (0.83–0.86)	0.89 (0.87–0.93)	0.03 (−0.008–0.09)	0.217
Total fat (%)	41.2 (41.1–43.8)	39.2 (38.5–41.5)	−1.9 (−2.3–−1.3)	41.3 (41.2–43.3)	42.9 (42.2–43.0)	0.1 (−0.4–1.0)	0.056
Trunk fat mass (kg)	18.3 (17.9–18.4)	16.7 (15.8–17.4)	−1.7 (−2.2–−0.9)	16.6 (14.6–18.2)	17.2 (16.4–18.2)	0.6 (0.3–0.6)	**0.016**
Endocrine							
Testosterone (nmol/L)	1.77 (1.20–1.83)	0.94 (0.74–1.22)	−0.46 (−0.78–−0.36)	1.29 (1.05–1.42)	0.96 (0.89–1.21)	−0.09 (−0.26–−0.03)	0.151
SHBG (nmol/L)	32.3 (23.0–38.2)	36.3 (28.5–38.3)	4.5 (4.0–5.5)	38.8 (32.3–38.8)	32.9 (31.6–42.9)	−1.1 (−5.4–4.1)	0.691
Metabolic							
HOMA-IR	2.9 (2.4–4.2)	1.5 (1.3–1.8)	−0.88 (−1.1–−0.001)	1.4 (1.1–1.6)	1.8 (1.7–2.2)	0.62 (0.40–0.79)	**0.032**
Adjusted adipocyte size (μm)	127 (124–129)	116 (113–125)	−2.2 (−11.3–2.2)	120 (116–122)	120 (114–120)	−0.44 (−3.1–6.9)	0.421

Values presented as median interquartile range (25th–75th). Significant results presented in bold. BMI, body mass index; HOMA-IR, homeostatic model assessment for insulin resistance; SHBG (sex hormone binding globulin); WHR, waist-hip ratio.

**Table 4 ijms-25-11566-t004:** Association between baseline variables and weight change in the whole sample (*n* = 55).

Variable	R^2^	*p*-Value
*GSTM5*	0.182	**0.0014**
WHR	0.162	**0.003**
FAI	0.145	**0.006**
SHBG (nmol/L)	0.135	**0.007**
*H3C2*	0.103	**0.02**

Simple linear regression analysis of baseline variables and change in weight (%). Significant results presented in bold. FAI, free androgen index; SHBG (sex hormone binding globulin); WHR, waist-hip ratio.

## Data Availability

The original contributions presented in the study are included in the article/Appendix A, further inquiries can be directed to the corresponding author.

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
