# Peer review of "Weight Changes Are Linked to Adipose Tissue Genes in Overweight Women with Polycystic Ovary Syndrome"

_ijms, 2024, doi:10.3390/ijms252111566_

Round 1

Reviewer 1 Report

Comments and Suggestions for Authors

The authors have addressed an interesting and important topic of the contribution of the genetic background of adipose tissue in obese women with PCOS syndrome. The presented research is structured, logical and clear. The study was designed in an appropriate manner. The authors have well described the study and control group. The study used modern, adequate molecular biology techniques. The results are presented in a clear, accessible and reliable manner. The results are not based on a large group of female patients, which is also understandable for conducting studies with lifestyle modification and outcome control. However, the presented analysis offers interesting perspectives and may provide a starting point for further research. Furthermore, it may give new insights to clinicians on the implementation of therapeutic interventions especially in obese women with PCOS, taking genetic analyses specifically into account. 

Author Response

Thank you for your positive comments.

Reviewer 2 Report

Comments and Suggestions for Authors

The paper entitled “Weight Changes Are Linked to Adipose Tissue Genes in Overweight Women with Polycystic Ovary Syndrome” is intriguing and, in my opinion, holds potential for publication in the International Journal of Molecular Sciences. It is well-established that life style changes and weight loss can improve the condition of PCOS, making the identification of specific genes influencing weight loss success a valuable area for research.

However, I do have some concerns and suggestions for improvement:

1.       The authors analyze gene expression, not the genes themselves, and this distinction should be clearly reflected throughout the manuscript.

2.       The Introduction lacks information on gene expression parameters  should be addressed.

3.       What was the rationale behind selecting these specific gene expressions for analysis? Particularly, why was the decision made to investigate genes associated with retinoblastoma, a condition not directly related to PCOS or metabolic function?

4.       The authors highlight the role of GST in oxidative stress and weight loss, but its primary function is in the second phase of biotransformation. There are other crucial enzymes involved in glutathione metabolism, such as glutathione peroxidase or glutathione reductase? Why were these not included in the study?

5.       The BMI cut-off for inclusion was >27, yet overweight classification begins at >25. Could the authors clarify why they chose this higher cut-off point?

6.       The authors report a significant decrease in FAI value and testosterone level in the minimal intervention group, with no significant changes in behavioral modification intervention group? How do the authors explain this findings, especially, when lifestyles are generally expected to improve PCOS-related conditions?

7.       In Figure 1, what does the symbol “**” represent? This should be clarified for the readers

8.       In Table 2, the p-value should be accompanied by an explanation of which specific comparisons is refers to?

9.       The study only includes women with PCOS, who meet all three criteria of the Rotterdam diagnosis. Therefore, the authors should specify that only phenotype 1 is represented in the study (“..and having all 3 of the Rotterdam diagnostic criteria of PCOS..” (line 254)

10.   The Discussion section mainly focuses on the results of GST expression, even though 40 genes were analyzed. This section should be expanded to explain why these 40 genes were chosen and to provide further discussion on the lack of significant results for many of them.

11.   In my opinion, the reported weight gain of 1.12% (0.19 – 2.04) over 4 months is relatively minor to categorize as  “the weight gain group”.  The lower end of 0.19 could be within the margin of error for the measurement equipment. The authors should be provide more details about the equipment and methods used for anthropometric assessments.

12.   Further clarification is needed regarding the source of the subcutaneous adipose tissue samples. Where the biopsies standardized across all participants?

13.    A more details description of the behavioral modification intervention and the minimal intervention is required. Specifically, what were the components of the interventions, and how were they implemented.

I look forward to seeing the revised version.

Author Response

Dear Reviewer,

Thank you for your valuable comments. We have changed the manuscript accordingly. Certain responses to the comments also have references, please see attached word file. 

Comment 1: The authors analyze gene expression, not the genes themselves, and this distinction should be clearly reflected throughout the manuscript.

Response:  This is now corrected.

Comment 2: The Introduction lacks information on gene expression parameters  should be addressed.

Response: We have added information about some relevant gene expression parameters in the Introduction.

Comment 3: What was the rationale behind selecting these specific gene expressions for analysis? Particularly, why was the decision made to investigate genes associated with retinoblastoma, a condition not directly related to PCOS or metabolic function?

Response: In this study, we explored the expression of genes involved in metabolic pathways of potential importance for weight changes after lifestyle intervention in overweight/obese women with PCOS. Expression of genes differing significantly between the subgroups of weight loss and weight gain in the microarray analysis were processed in a gene list enrichment analysis of metabolic pathways. In those pathways, retinoblastoma genes in cancer were included. Although genes involved in retinoblastoma are not relevant, we decided to validate all significant genes from the analysis of metabolic pathways. In the method section, we have now clarified that metabolic pathways were the focus in the gene list enrichment analysis.

Comment 4: The authors highlight the role of GST in oxidative stress and weight loss, but its primary function is in the second phase of biotransformation. There are other crucial enzymes involved in glutathione metabolism, such as glutathione peroxidase or glutathione reductase? Why were these not included in the study?

Response:  As described in comment #3, we only validated the significantly expressed genes involved in different metabolic pathways from the pathway analysis. Other genes involved in glutathione metabolism were not significantly expressed in the pathway analysis.

Comment 5: The BMI cut-off for inclusion was >27, yet overweight classification begins at >25. Could the authors clarify why they chose this higher cut-off point?

Response:  In previous papers, BMI <27 has been used as a cut-off for a non-obese PCOS population considered to have less risk for metabolic complications [1]. Consequently, we used BMI > 27 as a cut-off to investigate a population considered to have increased risk of metabolic complications.

Comment 6: The authors report a significant decrease in FAI value and testosterone level in the minimal intervention group, with no significant changes in behavioral modification intervention group? How do the authors explain this findings, especially, when lifestyles are generally expected to improve PCOS-related conditions?

Response:  We agree that this was unexpected. One explanation can be that the baseline value for testosterone and FAI were non-significantly higher in the minimal intervention group compared to the behavioral modification group. In the original publication we showed that there were significant reductions in FAI, fasting insulin and HOMA for the whole group of women at 12-month follow-up [2].

Comment 7: In Figure 1, what does the symbol “**” represent? This should be clarified for the readers

Response:  This is now corrected and explained.

Comment 8: In Table 2, the p-value should be accompanied by an explanation of which specific comparisons is refers to?

Response: The p-value refers to a comparison between change in the respective group, as denoted in the header.

Comment 9: The study only includes women with PCOS, who meet all three criteria of the Rotterdam diagnosis. Therefore, the authors should specify that only phenotype 1 is represented in the study (“..and having all 3 of the Rotterdam diagnostic criteria of PCOS..” (line 254)

Response:  Thank you for the comment. The manuscript has been revised accordingly. A new reference has also been added containing information about different phenotypes of PCOS.  

Comment 10: The Discussion section mainly focuses on the results of GST expression, even though 40 genes were analyzed. This section should be expanded to explain why these 40 genes were chosen and to provide further discussion on the lack of significant results for many of them.

Response:  Thank you for the comment, as described in comments #3 and #4, these genes were chosen based on microarray analysis of the two subgroups followed by pathway analysis focusing on metabolic pathways. However, not all gene expressions were significant in the PCR analysis of the whole group, which is now included as a limitation in the Discussion. This could at least partly be due to limited power to detect differences. GSTM5 was highlighted since its expression at baseline was the strongest predictor of weight change and even stronger than clinical variables.

Comment 11: In my opinion, the reported weight gain of 1.12% (0.19 – 2.04) over 4 months is relatively minor to categorize as  “the weight gain group”.  The lower end of 0.19 could be within the margin of error for the measurement equipment. The authors should be provide more details about the equipment and methods used for anthropometric assessments.

Response: Division into weight categories was based on measurement of body weight on the same electronic scale at the clinic for all women wearing underwear and a light hospital shirt. This has now been added in the Method section. Furthermore, the patients underwent DEXA scans, as described in the original publication by Oberg et al [2]. Body weight values measured by DEXA resulted in the same division of groups based on the 50th centile of weight change from baseline.

Comment 12: Further clarification is needed regarding the source of the subcutaneous adipose tissue samples. Where the biopsies standardized across all participants?

Response:  The procedure of collecting abdominal subcutaneous fat biopsies is now clarified in the Method section. The biopsy collection was standardized, and the same procedure was performed on all participants by two operators. Subcutaneous adipose tissue was collected from the abdomen at the level between the umbilicus and the iliac crest through a small incision in the skin under local anesthesia and snap frozen and stored at -80°C.

Comment 13: A more details description of the behavioral modification intervention and the minimal intervention is required. Specifically, what were the components of the interventions, and how were they implemented.

Response:  Additional information has been added to the description of the behavioral modification intervention and the minimal intervention in the Method section. In addition to this, reference is made to the original article, Oberg et al [2].

Reviewer 3 Report

Comments and Suggestions for Authors

I found the manuscript to be excellent and have only two suggestions: explanation of abbreviations in the abstract and adding to the discussion an attempt to explain the relationship between GSTM5, a gene involved in glutathione metabolism, and weight loss. In my opinion, there is also a lack of background information on diet and physical activity to make the observations and conclusions complete.

Author Response

Dear Reviewer,

Thank you for your valuable comments. We have changed the manuscript accordingly. Certain responses to the comments also have references, please see attached word file. 

I found the manuscript to be excellent and have only two suggestions: 

explanation of abbreviations in the abstract and adding to the discussion an attempt to explain the relationship between GSTM5, a gene involved in glutathione metabolism, and weight loss.

Response:  Thank you for the positive remarks. Abbreviations have now been added to the abstract. We have further evolved the discussion about potential mechanisms for anti-obesity effects of glutathione.

In my opinion, there is also a lack of background information on diet and physical activity to make the observations and conclusions complete.

Response:  We have added more information about the intervention requested by reviewer 2. Since this was a behavioral modification intervention, we did not analyze food intake or physical activity specifically. However, energy expenditure and step count were measured by an accelerometer as now included in the Method section but there were no significant changes as reported in the original paper by Oberg et al [2].
